# Association of CD206 Protein Expression with Immune Infiltration and Prognosis in Patients with Triple-Negative Breast Cancer

**DOI:** 10.3390/cancers14194829

**Published:** 2022-10-03

**Authors:** Angélique Bobrie, Océane Massol, Jeanne Ramos, Caroline Mollevi, Evelyne Lopez-Crapez, Nathalie Bonnefoy, Florence Boissière-Michot, William Jacot

**Affiliations:** 1Department of Medical Oncology, Institut Régional du Cancer de Montpellier (ICM), 34298 Montpellier, France; 2Institut de Recherche en Cancérologie de Montpellier (IRCM), Inserm U1194, Institut du Cancer Montpellier (ICM), 34298 Montpellier, France; 3Biometrics Unit, Institut Régional du Cancer de Montpellier (ICM), 34298 Montpellier, France; 4Translational Research Unit, Institut Régional du Cancer de Montpellier (ICM), 34298 Montpellier, France; 5Institut Desbrest d’Epidémiologie et de Santé Publique, IDESP UMR UA11 INSERM-Université de Montpellier, 34090 Montpellier, France; 6Department of Medicine, Montpellier University, 34298 Montpellier, France

**Keywords:** triple-negative breast cancer (TNBC), tumor-associated macrophages (TAM), prognosis, CD206, CD163, CD68, IRF8

## Abstract

**Simple Summary:**

Triple-negative breast cancers (TBNCs) represent 10–20% of all breast cancers. TNBCs are more frequent in younger women, and present more aggressive features and poorer prognosis. Few specific therapeutics are available for TNBC treatment and it is crucial to better characterize TNBC biology in order to discover new therapeutic targets. In this study, we focused on the immune tumor microenvironment, particularly on macrophages that have been less studied in the context of TNBC. Macrophages are very plastic cells; their phenotype and function can change depending upon environmental conditions. Therefore, we used four macrophage markers to quantify the macrophage infiltrate (CD68, IRF8, CD163, and CD206) in tumors from 285 patients with TNBC. We demonstrated for the first time that a population of macrophages, defined by CD206 expression, delineates a subgroup of TNBCs that may have a better prognosis. These results could help to refine the patients’ prognosis and develop new therapeutic strategies.

**Abstract:**

Background: Triple-negative breast cancers (TNBCs) have a worse prognosis, but might respond to immunotherapies. Macrophages are plastic cells that can adopt various phenotypes and functions. Although they are a major immune population in TNBCs, the relationship between tumor-associated macrophages (TAMs) and TNBC progression has been rarely explored, with controversial results. Methods: We evaluated the prognostic impact of TAMs, quantified by immunohistochemistry with anti-CD68, -IRF8, -CD163, and -CD206 antibodies, in a well-described cohort of 285 patients with non-metastatic TNBC. Results: CD68 (*p* = 0.008), IRF8 (*p* = 0.001), and CD163 (*p* < 0.001) expression positively correlated with higher tumor grade, while CD206 was associated with smaller tumor size (*p* < 0.001). All macrophage markers were associated with higher tumor-infiltrating lymphocyte numbers and PD-L1 expression. Univariate survival analyses reported a significant positive correlation between CD163^+^ or CD206^+^ TAMs and relapse-free survival (respectively: HR = 0.52 [0.28–0.97], *p* = 0.027, and HR = 0.51 [0.31–0.82], *p* = 0.005), and between CD206^+^ TAMs and overall survival (HR = 0.54 [0.35–0.83], *p* = 0.005). In multivariate analysis, there was a trend for an association between CD206^+^ TAMs and relapse-free survival (HR = 0.63 [0.33–1.04], *p* = 0.073). Conclusions: These data suggest that CD206 expression defines a TAM subpopulation potentially associated with favorable outcomes in patients with TNBC. CD206 expression might identify an immune TNBC subgroup with specific therapeutic options.

## 1. Introduction

Triple-negative breast cancers (TNBCs) represent 10–20% of all breast cancers. They are defined by a lack of estrogen receptor (ER) and progesterone receptor (PR) expression, and the absence of HER2 overexpression/amplification [1,2]. TNBCs have a specific clinical profile. Indeed, they are more frequent in younger women and a context of the *BRCA1* mutation, display more aggressive features, and are associated with a worse prognosis [1,3]. Specific targeted therapies are widely used for the other breast cancer types, but those currently available for TNBC treatment are restricted only to some TNBC subgroups [4,5,6,7].

In the past decades, much work has focused on characterizing TNBCs with the objective of finding therapeutic targets. Genomic-based approaches led to the description of TNBC molecular subtypes [2,8]. Perou et al. initially defined five clusters [1,2]. Later, Lehman et al. reported six subgroups based on unique gene expression profiles: basal-like 1, basal-like 2, immunomodulatory, mesenchymal-like, mesenchymal stem-like, and luminal androgen receptor (LAR) [8]. More recently, these groups were refined into four subgroups: basal-like 1, basal-like 2, mesenchymal, and LAR [9]. TNBC molecular subtypes correlate with distinct prognoses, various levels of sensitivity to chemotherapy [9], and are associated with different therapeutic targets [8].

Besides tumor biology, it was demonstrated that the tumor immune microenvironment plays a crucial role in cancer development [10], with therapeutic implications illustrated by the development of immune checkpoint inhibitors [11]. TNBCs have a distinct immune ecosystem compared to other breast cancers: (i) higher tumor mutational burden and higher neoantigen load [12], and consequently (ii) higher infiltration by immune cells, particularly by tumor-infiltrating lymphocytes (TILs) [13]. These characteristics are crucial for predicting the response to immunotherapy [14]. Recent studies on immune checkpoint inhibitors alone or in combination with chemotherapy or with other targeted therapies reported a benefit in patients with TNBC [15]. Very recently, the combination of immunotherapy with chemotherapy has been approved for neoadjuvant and first-line treatment of metastatic TNBC [16,17]. A better characterization of TNBC immune infiltrate may help to determine which patients might benefit from immunotherapy and identify factors of resistance that could constitute new therapeutic targets.

Various immune cell types from the lymphoid and myeloid lineages are present in the tumor microenvironment [18]. In TNBC, a higher level of infiltration by CD8^+^ cytotoxic T cells, CD4^+^ helper T cells, natural killer cells, and B lymphocytes has been associated with a better prognosis [18]. Interestingly, CD4^+^ FOXP3^+^ regulatory T cells also predict improved outcomes in these patients [19,20], unlike in other cancer types. Fewer data are available on the prognostic significance of myeloid cells in TNBC. Among myeloid cells, macrophages are the main immune population in most tumor types, including breast cancer, as a consequence of the important release of inflammatory signals [21]. Tumor-associated macrophages (TAMs) can present a large spectrum of phenotypes that share similarities with the “M2-phenotype” of the M1/M2 classification of macrophages [22]. TAMs are generally identified by the expression of CD68, a pan-macrophage marker, or/and CD163, a scavenger receptor that is upregulated in M2 macrophages. As TAMs can adopt various phenotypes [23], using only these two markers appears insufficient. CD206 (C-type mannose receptor 1) is also overexpressed on M2-macrophages [24], and has been detected on TAMs in several cancer types [25,26,27]. However, it has not been analyzed in TNBCs. To date, M1 markers have been rarely investigated in the tumor microenvironment. Interferon regulatory factor 8 (IRF8) is a transcription factor involved in the developmental program of the myeloid cell lineage, in antigen presentation by professional antigen-presenting cells, and in the production of pro-inflammatory cytokines that promote T-cell adaptive immune response [28]. IRF8 directly participates in the acquisition of M1 macrophage functions [29,30], and was recently used to characterize M1 TAMs in gastric cancer [26]. The role of TAMs in the TNBC microenvironment has been poorly investigated. The description of the immune ecosystem in breast tumors based on immune gene expression and immunofluorescence imaging revealed that the TNBC microenvironment is enriched in macrophages, compared with hormone receptor-positive tumors [31]. This result was confirmed by immunohistochemistry (IHC) studies [32,33]. Yet, data on TAM prognostic impact in patients with TNBC are limited and controversial [34,35,36].

Therefore, we quantified TAM infiltrate in a well-characterized and previously described cohort of 285 patients with TNBC [37,38,39] by assessing the expression of four macrophage markers (CD68, IRF8, CD163, and CD206) by IHC, and evaluated the impact of these TAM populations on overall survival (OS) and relapse-free survival (RFS).

## 2. Materials and Methods

### 2.1. Patients and Tumor Samples

Between 2002 and 2010, tumor samples from 1695 patients with breast cancer treated at the Montpellier Cancer Institute (ICM) were prospectively included in a dedicated tumor biobank (Biobank number BB-0033-00059). Patients had a unifocal, unilateral, non-metastatic disease, and tumors were resected before any systemic treatment. Tumors were considered hormone receptor-negative when the expression of ER and PR was detected in <10% of tumor cells by IHC. HER2 status was determined by an IHC-based evaluation of HER2 protein expression using the A485 monoclonal antibody (Dako/Agilent, Santa Clara, CA, USA). Tumors with scores 0 and 1+ were considered HER2-negative. Tumors with a score of 2+ were also considered HER2-negative when no gene amplification was detected by fluorescence or chromogenic in situ hybridization. This allowed the identification of 418 TNBCs. Their biological parameters were described previously: cytokeratin (CK) 5/6, epithelial growth factor receptor (EGFR), programmed death ligand 1 (PD-L1), PD-1, androgen receptor (AR) and forkhead box protein A1 (FOXA1) expression, and TIL quantification [38,39,40]. The basal-like phenotype was defined by the expression of CK5/6 and/or EGFR in >10% of tumor cells by IHC. The molecular apocrine phenotype was defined by the nuclear expression of both AR and FOXA1 in at least 1% of tumor cells. For the present study, 285 patients with TNBC from this cohort were selected because they had tumor samples in which the expression of macrophage markers could be evaluated by IHC (Figure 1). Patients were treated in accordance with our institution guidelines [41]. The median follow-up was 10.15 years.

The protocol of this study was reviewed and approved by the Montpellier Cancer Institute Institutional Review Board (ICM-CORT-2019-30). All included patients provided written informed consent.

### 2.2. Tissue Microarrays (TMA) and Immunohistochemistry

Macrophage infiltrate was assessed on the same TMA blocks used in our previous studies on this cohort [38,39,42,43]. For each tumor, the area of invasive carcinoma was identified on a Hematoxylin-Eosin-Saffron (HES) section. Using the corresponding formalin-fixed paraffin-embedded (FFPE) tissue blocks, two 1-mm cores were selected within this area, sampled, and placed at specific coordinates using the Manual Tissue Arrayer device (Beecher Instruments, Sun Prairie, WI, USA). Six TMAs were produced (*n* = 349 samples in total). Four 3 µm-thick serial sections were cut from each TMA and used for IHC. All slides were processed on a Dako-Link platform, first using the PT-Link module for simultaneous dewaxing and antigenic retrieval, and then with the Dako-Link48 Autostainer and FLEX+ visualization system (Dako/Agilent, Santa Clara, CA, USA) for all the immunostaining steps. Following antigen unmasking in Target Retrieval Low (CD206) or High (CD68, CD163, IRF8) pH solution and blocking of endogenous peroxidase activity with the EnVision FLEX Peroxidase Block Solution (Dako/Agilent, Santa Clara, CA, USA), sections were incubated with mouse monoclonal antibodies against CD68 (ready to use, clone PG-M1, Dako/Agilent, Santa Clara, CA, USA), CD163 (ready to use, clone 10D6, BioSB, Santa Barbara, CA, USA), ICSBP/IRF8 (1:500 dilution, clone E-9, Santa Cruz Biotechnologies, Dallas, TX, USA), or rabbit polyclonal antibody against CD206 (1:5000 dilution, ab64693, Abcam, Cambridge, UK). After two rinses in EnVision FLEX wash buffer, slides were incubated with a horseradish peroxidase-labeled polymer coupled to secondary anti-mouse and anti-rabbit antibodies (Envision FLEX HRP, Dako), followed by incubation with EnVision FLEX Substrate Working Solution containing 3,3′-diaminobenzidine as the chromogen (Dako) at room temperature for 10 min. Sections were counterstained with EnVision FLEX Hematoxylin (Dako), rinsed in tap water for 5 min, dehydrated, and mounted with a coverslip. The detailed IHC procedures of other IHC markers used in this study (ER, PR, EGFR, CK5/6, AR, and FOXA1) are described in previous studies published on the same cohort of patients [38,39,42,43,44]. For PD-L1 detection, the anti-PD-L1 rabbit monoclonal antibody (clone SP142, Roche Diagnostics, Basel, Switzerland) and the Autostainer Link48 platform (Dako/Agilent, Santa Clara, CA, USA) were used, followed by the Flex^®^ system for signal amplification and diaminobenzidine tetrahydrochloride as the chromogen.

### 2.3. Analysis of TAM Marker Expression

Stained sections were digitalized with the NanoZoomer slide scanner system (Hamamatsu Photonics, Hamamatsu City, Shizuoka Pref., Japan) and a ×20 objective. Sections were analyzed independently by two trained observers (Bobrie A. and Ramos J.) blinded to the patients’ clinicopathological characteristics. Discordant results between observers were examined again together to reach consensus. Two methods of quantification were chosen as more reliable in the function of the expression level of each marker: markers expressed in a high percentage of cells (CD68 and CD163) were assessed semi-quantitatively, while markers expressed in a limited number of cells (CD206 and IRF8) were evaluated by absolute quantification (number of cells/mm^2^). The expression levels of CD68 and CD163 were assessed using a four-score scale ranging from no/very low (score 0; meaning no or rare positive cells), weak (score 1; less than 20% of positive cells), moderate (score 2; 20 to 50% of positive cells), to very high expression (score 3; ≥50% of positive cells). The percentage of positive cells was defined as the number of positive cells divided by the total number of stromal and tumor cells inside the invasive tumor-containing areas in each spot. The absolute number of CD206^+^ and IRF8^+^ cells was quantified, and their densities were reported as the number of positive cells per mm^2^. The mean value of duplicate experiments was calculated for each patient and each marker. TMA cores that were missing, containing fewer than 10 cancer cells, or with significant artefacts were not scored. Finally, CD68, CD163, CD206, and IRF8 expression could be evaluated in 267, 276, 272, and 277 tumors, respectively.

To analyze the clinicopathological significance of each TAM marker, two groups (“low” and “high” expression) were compared. For CD206 and IRF8, the median expression value was chosen. This is the most common, unbiased, and objective way to delineate two groups, while optimizing the statistical power of the analysis by balancing the two analyzed groups. For CD68 and CD163, three groups seemed to be more suitable initially to homogeneously divide the population. However, this classification did not bring any additional or more relevant information and made the results more difficult to follow. Therefore, two groups with “low” (score ≤ 2) and “high” (score > 2) expression were then used.

### 2.4. TIL Assessment

TILs were evaluated on HES-stained digitalized TMA sections by a trained pathologist, according to the International TIL Working Group guidelines [45]. As recommended, only stromal TILs were quantified, while TILs within the tumor nest were not taken into account. Besides TIL quantification, expression of CD3 and CD8 was evaluated by IHC (number of immunoreactive cells/mm^2^) and image analysis, as previously described [42].

### 2.5. Statistical Analysis

Quantitative variables were described as the number of observations, medians, minimum and maximum values. Qualitative variables were described as the number of observations and frequency of each modality. Data were compared with Pearson’s chi-square or Fisher’s exact test when the theoretical numbers were <5. OS was defined as the time between the date of surgery and the date of death, whatever the cause. Patients alive or lost to follow-up were censored at the date of the last news. RFS was defined as the time between the date of surgery and the date of recurrence. Patients living without recurrence and patients lost to follow-up were censored at the date of the last news. Patients who died before any recurrence were censored at the date of death. The Kaplan–Meier method was used to analyze survival data and estimate the median survival rates and times. Survival distributions were compared using the log-rank test. Multivariate analyses were performed using the Cox proportional hazard model to estimate hazard ratios (HR) and their 95% confidence intervals (95% CI). All variables with *p* < 0.15 in univariate analysis were selected for multivariate analysis and a backward selection procedure was performed. Statistical analyses were performed with STATA 16.0 (StatCorp, College Station, TX, USA).

## 3. Results

### 3.1. Patient and Tumor Characteristics

Tumor infiltration by macrophages was evaluated in 285 TNBC samples from previously well-characterized TMAs [38,39,42,43] (Figure 1). The clinicopathological characteristics of these 285 patients with TNBC are presented in Table 1. Their median age was 57.8 years, and 213 (75%) received adjuvant chemotherapy. None of them received targeted therapy or any investigational product. The main histological type was ductal carcinoma (83.7%). Tumor size at diagnosis was T1 and T2 in 44.6% and 49.1% of patients, respectively. Initial lymph node invasion by tumor cells was detected in 102 patients (35.8%), and 77.0% of tumors were classified as histological grade 3. IHC analysis indicated that 63.6% of TNBC samples had a basal-like phenotype (CK5/6 and EGFR expression > 10%), and 42.2% had a molecular apocrine phenotype (AR and FOXA1 expression ≥ 1%). TIL percentage and PD-L1 expression on tumor or stromal cells are presented in Table 1.

### 3.2. TAM Characterization and Quantification

To assess the prognostic impact of different macrophage populations in the TNBC microenvironment, the expression of a pan-macrophage marker (CD68), a M1 macrophage marker (IRF8), and two M2-like macrophage markers (CD163 and CD206) was evaluated by IHC (Appendix A). The quantification was performed in the whole tumor area of each TMA spot.

CD68 and CD163 presented a strong cytoplasmic signal in cells with macrophage morphology (Figure 2A,B and Appendix A). They were detected in 265/267 (99.3%) and 268/276 (97.1%) of all evaluable samples, respectively, with a higher proportion of score 2 (i.e., moderate expression) tumors (Appendix A). For each marker, samples were divided into two groups (Appendix A): low (scores from 0 to 2) (*n* = 173 for CD68, 64.8%, and *n* = 200 for CD163, 72.5%) and high expression (score > 2; *n* = 94 for CD68, 35.2%, and *n* = 76 for CD163, 27.5%). CD206 signal was also cytoplasmic in macrophages (Figure 2C, Appendix A), and was only rarely detected in endothelial cells, cells with a neutrophil morphology, or tumor cells. The transcription factor IRF8 presented a nuclear signal, as expected (Figure 2D, Appendix A), in cells with macrophage morphology. A weak signal was rarely observed in tumor cells. CD206 and IRF8 were expressed in 199/272 (73.2%) and 156/277 (56.3%) of the samples, respectively. Quantification of CD206^+^ and IRF8^+^ cell density showed a high range of infiltration (Appendix A): from 0 to 441.2 cells/mm^2^ for CD206 with a median of 7.90 cells/mm^2^, and from 0 to 387.1 cells/mm^2^ for IRF8 with a median of 3.11 cells/mm^2^. The median density values were used as an objective cut-off to classify samples into two groups (low and high density).

As multiplexed staining was not performed, it was not possible to assess marker co-expression in macrophages. Nevertheless, IHC was performed in consecutive sections, and visual analysis suggested similar CD68 and CD163 expression profiles in the same tumor sample, although CD68 seemed to be expressed in a larger proportion of macrophages. Conversely, CD206 and IRF8 were detected in fewer cells in the tumor microenvironment. The expression levels of the four TAM markers were significantly and positively correlated (Appendix A), although CD206 expression was less correlated with CD68 and CD163 expression.

### 3.3. Association of TAM Markers with TNBC Clinicopathological Features

CD68 (pan-macrophage marker) expression (Table 2) was significantly associated with younger age (*p* = 0.020) and worse histological grade (*p* = 0.008). High IRF8 and CD163 (but not CD206) expression levels were also correlated with a higher histological grade (*p* = 0.001 for both). Only CD206 was correlated with tumor size: higher CD206^+^ TAM density was found in smaller tumors (*p* < 0.001). The four TAM markers were not associated with the tumor nodal status.

The molecular apocrine TNBC phenotype was correlated with lower CD68 (*p* < 0.001) and CD163 (*p* = 0.001) expression, while the basal-like phenotype was associated with higher CD163 expression (*p* = 0.001). There was no significant correlation with CD206 or IRF8 expression.

All four TAM markers were significantly and positively associated with higher TIL density (*p* < 0.001), higher PD-L1 expression in tumor (*p* < 0.001) and stromal cells (*p* < 0.001 for CD68, IRF8 and CD163; *p* = 0.002 for CD206). 

### 3.4. Survival Analysis

The median follow-up was 10.15 years (95% CI [9.3–10.7]). The 10-year RFS rate was 74% (95% CI [68.0–79.0]) with a 10-year OS rate of 67% (95% CI [60.0–72.0]).

In univariate analysis (Table 3), classical TNBC prognostic variables and higher TIL density were associated with better RFS and OS. A higher percentage of PD-L1^+^ cells in the tumor tended to predict better RFS, but did not reach the significance threshold (*p* = 0.055). There was no correlation between survival parameters and CD68 or IRF8 expression. Conversely, high expression of CD163 and CD206 was associated with better RFS (HR = 0.52; 95% CI [0.28–0.97] and HR = 0.51; 95% CI [0.31–0.82], respectively) (Figure 3). Only CD206 expression was significantly correlated with better OS (HR = 0.54; 95% CI [0.35–0.83], *p* = 0.08) (Figure 3). Then, survival (RFS and OS) was analyzed in function of the proportion of M1/M2 TAMs according to their IRF8 and CD206 expression levels. Four groups were defined: IRF8^Lo^/CD206^Lo^, IRF8^Lo^/CD206^Hi^, IRF8^Hi^/CD206^Lo^, and IRF8^Hi^/CD206^Hi^. Including IRF8 expression level did not bring any more information because the curves were similar for the two CD206^Lo^ groups (with worse survival) and the two CD206^Hi^ groups (with better survival), whatever their IRF8 status.

To determine whether the correlation between CD206 expression and survival parameters might be explained by its association with smaller tumor size or higher TIL percentage, subgroup analyses were performed. The prognostic impact of CD206 expression was more pronounced in larger tumors (Appendix A) and tumors less infiltrated by lymphocytes (Appendix A).

The statistically significant variables in multivariate analysis are reported in Table 4. Lymph node positivity correlated with worse RFS and OS, and higher tumor size with worse OS. Conversely, adjuvant chemotherapy and TILs > 5% remained associated with better RFS and OS, and ductal histology with better OS. A trend for an association between high CD206 expression and better RFS was observed (HR = 0.63; 95% CI [0.33–1.04], *p* = 0.073).

## 4. Discussion

TNBCs are aggressive tumors with poor outcomes and few targeted treatment options. Much ongoing research is focused on improving their management. TNBCs are good candidates for immunotherapy [46]. Very recently, immune checkpoint inhibitors in combination with chemotherapy have been approved for the neoadjuvant treatment of stage II and III TNBCs and as first-line treatment of metastatic TNBCs [16,17]. Besides these settings, the clinical impact of immune checkpoint inhibitors remains limited [15], possibly due to the presence of immunosuppressive immune cells in the tumor microenvironment. Indeed, the immune infiltration profile is different in TNBCs compared with other breast cancer types [31], including macrophage enrichment in the tumor microenvironment. TAMs can be pro-tumorigenic or immunosuppressive and can interfere with some cancer therapies, such as immunotherapy [47]. TAMs correlate with poor prognosis in most cancer types, including breast cancer; however, few and contradictory data are available on TNBCs [34,48,49]. In the present study, we characterized macrophage infiltration and investigated its impact on survival in a cohort of 285 patients with TNBC.

As a single marker cannot accurately differentiate the M1 and M2 phenotypes, we decided to use four macrophage markers to determine whether one of them could be of particular interest in this TNBC cohort. Although the classical IHC method used in this study precluded the evaluation of their co-expression, we could compare expression patterns on serial sections. We noted that CD68 and CD163 presented a similar strong expression profile, while CD206 and IRF8 were expressed by more restricted TAM populations. Therefore, we chose different methods to quantify these two marker groups. As the high percentage of CD68- and CD163-positive cells precluded a reliable manual count, we opted for a semi-quantitative scoring method. Conversely, semi-quantitative scores were not precise enough to discriminate CD206 and IRF8 expression profiles due to the low number of positive cells. Therefore, we counted the absolute number of CD206- and IRF8-positive cells by mm^2^. Although using two methods of quantification induced heterogeneity in result reporting, this option was the most suitable for each individual marker, to precisely describe the expression level of each marker. Some studies reported CD163 and CD68 co-localization in macrophages [26], including in breast cancer [36], in agreement with our observations. This can be explained by the high proportion of M2-type TAMs in the tumor, or by a lack of CD163 specificity. Indeed, although CD163 is upregulated in M2 macrophages, it can be expressed by other macrophage populations [50]. In our study, CD206 expression was less correlated with CD68 and CD163 expression. Two hypotheses might explain this result, but co-staining will be needed to conclude. First, we could have detected a CD68^−^CD163^−^CD206^+^ cell population made of immature myeloid cells or tissue-resident macrophages. It has been reported that tissue-resident macrophages proliferate in the context of cancer, but little is known about their role in tumor progression [51,52]. Second, we could have identified a more restricted population of CD68^+^CD163^+^CD206^+^ M2 TAMs in a tumor microenvironment that is relatively less infiltrated by other CD68^+^CD163^+^ TAMs. These results highlight the lack of specificity of most of the markers currently used to assess TAM composition. Nevertheless, CD206^+^ cell infiltration appears to be distinct from the cell subpopulations identified by the other TAM markers.

We found that the global macrophage infiltrate (i.e., CD68^+^ cells) in this TNBC cohort correlated with worse histological grade, as previously described [53], but not with clinical outcome. Published results are discordant on this topic. CD68^+^ TAM infiltration was an independent factor of worse survival in a cohort of 287 patients with TNBC [34], and was associated with a poor outcome in another study on 200 basal-like tumors [48]. Conversely, no significant correlation was found in two cohorts of 107 and 96 patients with TNBC [35,54]. CD68 is a pan-macrophage marker that does not discriminate among the different macrophage subpopulations, and this may explain these differences. Moreover, TAM spatial distribution (tumor nest versus tumor stroma) could be important [33,48,54,55]. In our study, due to the limited tumor area analyzed on TMA spots, we decided to quantify the macrophage infiltrate in the whole tumor area, to reduce the possible sampling bias.

We analyzed the prognostic significance of M1 TAMs by exploring IRF8 expression. Few studies evaluated the role of IRF8^+^ TAMs [26,56,57]. In our cohort, high IRF8^+^ cell density correlated with worse histological grade, but not with survival parameters. Huang et al. found that in gastric cancer [26], CD68^+^IRF8^+^ TAMs are associated with an inflammatory environment rich in cell death signals. However, their number was lower compared with M2 TAMs and they did not correlate with survival. This is concordant with our observations. Moreover, the poor representation of IRF8^+^ TAMs in our cohort may explain the absence of significant association with the clinical outcome.

M2-like TAMs are usually characterized by CD163 expression. However, this marker has some limitations [50], and we decided to also include CD206 [24]. CD206 is expressed in TAMs in several cancer types [25,26,27,58]. To our knowledge, this is the first report that analyzed the clinical relevance of CD206^+^ TAMs specifically in the TNBC subtype. These two markers were related to different tumor clinicopathological parameters: CD163^+^ TAMs correlated with higher tumor grade and with basal-like phenotype, while CD206^+^ TAMs were associated with smaller tumor size, but not with TNBC molecular subtypes. Therefore, our results suggest that CD206 identifies a subpopulation of M2-like TAMs, as reported in publications using multiplexed analysis in other tumor types [25,26,58]. In the univariate survival analysis, increased CD163^+^ and CD206^+^ TAM infiltration correlated with better RFS (HR = 0.52; 95% CI [0.28–0.97], *p* = 0.027; HR = 0.51; 95% CI [0.31–0.82], *p* = 0.005), and CD206^+^ TAMs correlated also with better OS (HR = 0.54; 95% CI [0.35–0.83], *p* = 0.005). The association between CD206^+^ TAMs and RFS did not remain significant in multivariate analysis (HR = 0.63; 95% CI [0.33–1.04], *p* = 0.073). Considering the long median follow-up (10.15 years), and the fact that in our cohort, many deaths were not linked to cancer, we think that RFS better reflects the risk of relapse. Therefore, we conclude that both M2 markers are related to improved prognosis in patients with TNBC, particularly CD206, which identified a more restricted TAM subpopulation. CD206 expression also correlated with smaller tumor size and higher TILs. These two features predict a better prognosis, as confirmed by our analysis. Therefore, we asked whether these correlations might explain CD206 prognostic significance. We performed a subgroup analysis and found that the association between higher CD206 expression level and a better prognosis was more relevant in larger tumors and tumors with fewer TILs, removing the risk of a passive correlation between the absolute quantity of these variables and the results obtained for CD206. This indicates that CD206 prognostic significance is not due to its correlation with smaller tumor size or with a higher TIL level.

This result was unexpected due to the TAM pro-tumoral functions described in most cancer types [59,60] and also in breast cancer [53,55,61], although some exceptions exist [62]. Few and controversial data are available for TNBCs. In 2018, Yang et al. found that CD163^+^ TAMs in the tumor stroma were independently associated with shorter OS and disease-free survival in a cohort of 200 patients with basal-like tumors [48]. However, the definitions of basal-like and TNBC are not strictly overlapping. In our cohort, high CD163^+^ TAMs were associated with basal-like tumors (using IHC surrogate markers), but we cannot conclude about their prognostic significance in this specific subgroup. More recently, Jamiyan et al. reported that CD163^+^ TAMs, detected by IHC, were an independent prognostic factor of worse RFS and OS in a cohort of 107 patients with TNBC [35]. Conversely, Pelekanou et al. found that CD163^+^ TAMs, quantified by multiplexed immunofluorescence analysis, were associated with a better prognosis in 160 patients with TNBC [36]. CD206 expression has not been studied in TNBCs, and results published in other breast cancer types are controversial. Koru-Sengul et al. found an association between CD206^+^ TAMs and worse RFS by univariate analysis in 150 patients [63]. Recently, Strack et al. described a subpopulation of CD206^+^ TAMs that correlated with better prognosis in two independent cohorts of 154 and 118 patients with breast cancer [25]. Our study highlighted, for the first time, a positive correlation between the CD206^+^ TAM subpopulation and better prognosis in TNBCs (univariate analysis). As CD206^+^ TAMs were significantly associated with smaller tumors in our series, this TAM population may contribute to limiting tumor progression. Gruosso et al. [64] proposed to delineate TNBC subtypes, based on their immune infiltrate profile. These subtypes poorly correlate with those defined by Lehman et al., based on tumor cell biology [8]. Discrepancies about the role of TAMs in TNBC might be related to these different immune microenvironments because local conditions influence TAM phenotype and function. CD206^+^ macrophage-enriched tumors had a specific macrophage infiltrate with proportionally fewer CD68^+^ and CD163^+^ TAMs. All TAM markers positively correlated with TIL level and PD-L1 expression in tumor and stromal cells. It has been shown that some TAMs express PD-L1 [65], but their clinical significance is still unclear [66,67,68,69]. It would be interesting to determine whether CD206^+^ TAMs also express PD-L1 because it is a therapeutic target, although, in our study, PD-L1 expression in stromal cells did not correlate with survival parameters. Other markers of interest, particularly with the aim of developing targeting strategies, should be analyzed, for instance, GRP94 [70]. To conclude, in our work, CD206 defines a distinct immune TNBC subgroup, based on a specific TAM infiltrate, that may correlate with a better prognosis.

Our study has some limitations. First, although the analysis concerned 285 patients, the population might have been too small to detect significant correlations between CD206^+^ TAMs and survival parameters in multivariate analysis. Second, the analysis concerned the whole tumor area without differentiating macrophages located in the tumor stroma and tumor nest, although it has been shown that intra-tumoral TAM localization might influence prognosis [33,48,54,55]. However, this differentiation could be associated with significant bias in the TMA context where small tumor areas are analyzed. Third, multiplexed analyses could not be performed to better define the spatial expression of the specific CD206^+^ macrophage subpopulation relative to the other markers. However, each TAM marker was assessed using serial sections, which allowed visual comparison of the global expression profile. To understand the role played by CD206^+^ TAMs in TNBCs, their phenotype and functions must be characterized using multiplexed imaging, such as sequential IHC staining and image stacking that allow for assessing 12 different markers in a single tumor slice [71]. Mass cytometry is another technology that could be used to detect a large number of biological markers and their spatial localization in the tumor environment [72].

## 5. Conclusions

We demonstrated, for the first time, in a large cohort of patients with TNBC that a TAM subpopulation, defined by CD206 expression, tends to correlate with improved prognosis. More investigations are necessary to better characterize this population, using multiplexed imaging or gene expression analysis, and to understand the underlying mechanisms (direct effect on tumor cells or modulation of the immune microenvironment). Multiparametric and spatial analyses of the immune microenvironment in CD206^+^ TAM-enriched TNBCs would help to understand the cross-talk of macrophages with other immune populations and tumor cells, with the ultimate aim of developing new therapeutic strategies.

## Figures and Tables

**Figure 1 cancers-14-04829-f001:**
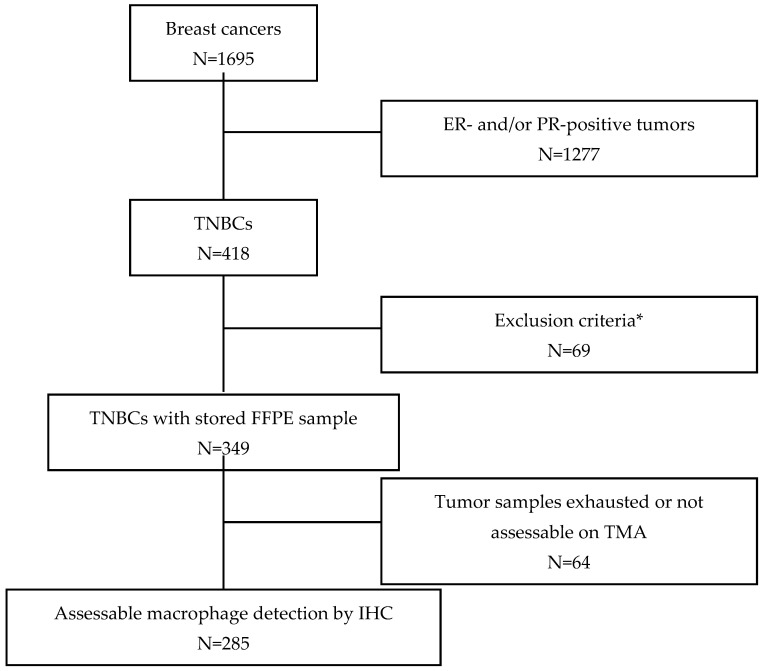
Consort diagram. * Exclusion criteria: Metastatic disease at time of tumor sampling, neoadjuvant treatment before tumor sampling, history of another invasive cancer in the previous 5 years, multifocal tumors.

**Figure 2 cancers-14-04829-f002:**
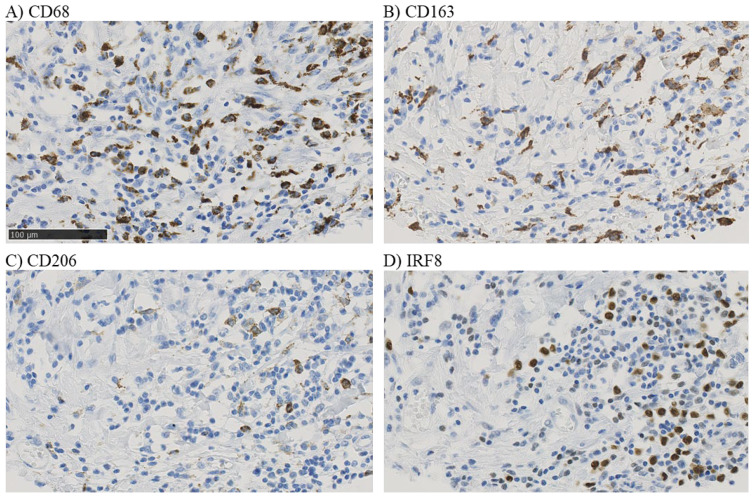
Example of CD68 (**A**), CD163 (**B**), CD206 (**C**), and IRF8 (**D**) immunostaining in serial sections of a single tumor from a patient, with × 40 magnification of a selected area (whole area in Appendix A).

**Figure 3 cancers-14-04829-f003:**
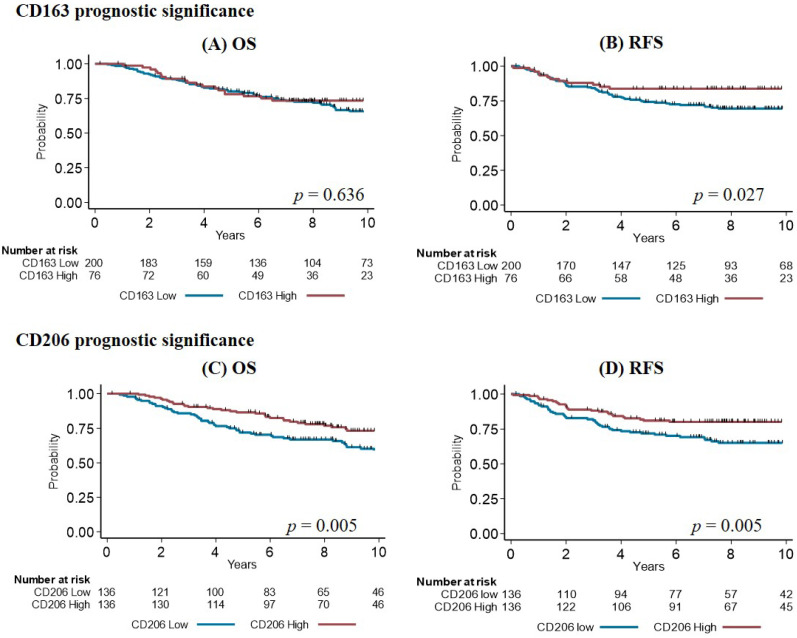
Overall survival (OS) (**A**,**C**) and relapse-free survival (RFS) (**B**,**D**) in function of CD163 (**A**,**B**) and CD206 (**C**,**D**) expression levels in TAMs.

**Table 1 cancers-14-04829-t001:** Patient and tumor characteristics.

Variable	Number of Patients (*n* = 285)	%
**Age (years),** median [min–max]	57.76	[28.54–89.10]
<55	126	44.21
≥55	159	55.79
**Tumor size**		
T1	127	44.56
T2	140	49.12
T3/T4	18	6.32
**Nodal status**		
N−	183	64.21
N+	102	35.79
**Histological grade**	3 missing values	
1–2	65	23.05
3	217	76.95
**Histology**	3 missing values	
Ductal	236	83.69
Lobular	15	5.32
Other	31	10.99
**Adjuvant chemotherapy**	1 missing values	
No	71	25.00
Yes	213	75.00
**Basal-like phenotype**	2 missing values	
No (≤10%)	103	36.40
Yes	180	63.60
**Molecular apocrine phenotype**	15 missing values	
No (<1%)	156	57.78
Yes (≥1%)	114	42.22
**TILs**	5 missing values	
≤5%	174	62.14
>5%	106	37.86
**PD-L1^+^ tumor cells**	22 missing values	
<1%	118	44.87
≥1%	145	55.13
**PD-L1^+^ stromal cells**	25 missing values	
0	45	17.31
[0–10]	86	33.07
[10–50]	71	27.31
≥50	58	22.31

Basal-like tumors were defined by CK5/6 and/or EGFR expression by IHC (>10% of tumor cells). Molecular apocrine tumors were defined by AR and FOXA1 positivity by IHC (≥1% of tumor cells); TILs: tumor-infiltrating lymphocytes according to the Salgado guidelines [45].

**Table 2 cancers-14-04829-t002:** Correlations between macrophage infiltrate and clinicopathological parameters of the TNBC cohort. The basal-like phenotype was defined by CK5/6 and/or EGFR positivity by IHC (>10% of tumor cells). The molecular apocrine phenotype was defined by AR and FOXA1 positivity by IHC (≥1% of tumor cells). The bold: the *p*-value was significant.

	CD68					IRF8					CD163					CD206				
	Low		High		*p*-Value	Low		High		*p*-Value	Low		High		*p*-Value	Low		High		*p*-Value
	*N*	%	*N*	%		*N*	%	*N*	%		*N*	%	*N*	%		*N*	%	*N*	%	
**Age (years)**																				
<55	70	40.46	52	55.32	**0.020**	47	38.84	78	50.0	0.064	84	42.0	40	52.63	0.113	54	39.71	65	47.79	0.179
≥55	103	59.54	42	44.68		74	61.16	78	50.0		116	58.0	36	47.37		82	60.29	71	52.21	
**Tumor size**																				
T1	77	44.51	44	46.81	0.364	62	44.60	61	44.20	0.716	89	44.50	36	47.37	0.695	45	33.09	79	58.09	**<0.001**
T2	83	47.98	47	50.00		68	48.92	71	51.45		97	48.50	37	48.68		77	56.62	53	38.97	
T3/T4	13	7.51	3	3.19		9	6.47	6	4.35		14	7.00	3	3.95		14	10.29	4	2.94	
**Nodal status**																				
N−	107	61.85	63	67.02	0.401	85	61.15	94	68.12	0.225	125	62.50	52	68.42	0.360	82	60.29	93	68.38	0.164
N+	66	38.15	31	32.98		54	38.85	44	31.88		75	37.50	24	31.58		54	39.71	43	31.62	
**Histological grade**																				
1–2	49	28.49	13	13.98	**0.008**	43	31.39	20	14.60	**0.001**	58	29.29	5	6.67	**<0.001**	30	22.56	33	24.26	0.741
3	123	71.51	80	86.02		94	68.61	117	85.40		140	70.71	70	93.33		103	77.44	103	75.74	
**Basal-like**																				
No (≤10%)	66	38.37	30	32.26	0.323	52	37.68	47	34.31	0.560	84	42.21	15	20.00	**0.001**	44	32.59	52	38.52	0.309
Yes	106	61.63	63	67.74		86	62.32	90	65.69		115	57.79	60	80.00		91	67.41	83	61.48	
**Molecular apocrine**																				
No (<1%)	81	50.31	65	71.43	**<0.001**	71	53.79	80	60.61	0.263	97	51.60	57	77.03	**<0.001**	76	58.46	76	58.91	0.941
Yes (≥1%)	80	49.69	26	28.57		61	46.21	52	39.39		91	48.40	17	22.97		54	41.54	53	41.09	
**TILs**																				
≤5%	132	77.19	29	31.87	**<0.001**	110	79.71	58	42.96	**<0.001**	154	77.39	15	20.83	**<0.001**	99	74.44	66	49.25	**<0.001**
>5%	39	22.81	62	68.13		28	20.29	77	57.04		45	22.61	57	79.17		34	25.56	68	50.75	
**PD-L1 tumor cells**																				
<1%	88	56.77	22	23.91	**<0.001**	76	59.84	39	30.00	**<0.001**	98	54.14	18	24.32	**<0.001**	68	54.40	42	33.07	**0.001**
≥1%	67	43.23	70	76.09		51	40.16	91	70.00		83	45.86	56	75.68		57	45.60	85	66.93	
**PD-L1 stromal cells**																				
0	33	21.43	9	10.00	**<0.001**	26	20.47	18	14.06	**<0.001**	37	20.55	8	11.11	**0.001**	27	21.95	15	11.90	**0.002**
[0–10]	58	37.66	23	25.56		60	47.25	25	19.53		70	38.89	14	19.45		47	38.21	34	26.99	
[10–50]	38	24.68	27	30.00		27	21.26	41	32.03		41	22.78	26	36.11		32	26.02	38	30.16	
≥50	25	16.23	31	34.44		14	11.02	44	34.38		32	17.78	24	33.33		17	13.82	39	30.95	

**Table 3 cancers-14-04829-t003:** Univariate analysis. Basal-like tumors were defined by CK5/6 and/or EGFR positivity by IHC (>10% of tumor cells). Molecular apocrine tumors were defined by AR and FOXA1 positivity by IHC (≥1% of tumor cells). The bold: the *p*-value was significant.

Variables	OS			RFS		
	HR	95% CI	*p*-Value	HR	95% CI	*p*-Value
**Age (years)**			**<0.001**			0.067
<55	1			1		
≥55	2.10	1.33–3.31		1.55	0.96–2.51	
**Tumor size**			**<0.001**			**<0.001**
T1	1			1		
T2/T3/T4	2.78	1.71–4.50		2.44	1.46–4.09	
**Nodal status**			**<0.001**			**<0.001**
N−	1			1		
N+	2.45	1.61–3.72		4.61	2.82–7.51	
**Histological grade**			0.472			0.904
1–2	1			1		
3	0.84	0.52–1.34		1.03	0.60–1.78	
**Histology**			**0.032**			0.600
Ductal	1			1		
Other	0.50	0.25–1.00		0.84	0.44–1.61	
**Adjuvant chemotherapy**			**<0.001**			**0.002**
No	1			1		
Yes	0.34	0.22–0.51		0.46	0.29–0.73	
**Basal-like phenotype**			0.697			0.550
No (≤10%)	1			1		
Yes	1.09	0.70–1.69		0.87	0.54–1.39	
**Molecular apocrine**						
No (<1%)	1		**0.041**	1		**0.032**
Yes (≥1%)	1.56	1.02–2.39		1.67	1.04–2.66	
**TILs**						
≤5%	1		**0.005**	1		**0.001**
>5%	0.51	0.32–0.83		0.42	0.24–0.74	
**PD-L1 tumor cells**			0.090			0.055
<1%	1			1		
≥1%	0.69	0.45–1.06		0.63	0.39–1.01	
**PD-L1 stromal cells**						
0	1			1		
[0–10]	1.42	0.75–2.69	0.191	1.33	0.68–2.61	0.069
[10–50]	0.85	0.42–1.74		0.56	0.25–1.26	
≥50	0.82	0.38–1.74		0.81	0.37–1.79	
**CD68**						
Low	1		0.852	1		0.299
High	0.96	0.61–1.50		0.77	0.46–1.28	
**IRF8**						
Low	1		0.495	1		0.456
High	0.86	0.56–1.32		0.84	0.52–1.34	
**CD163**						
Low	1			1		
High	0.89	0.54–1.46	0.636	0.52	0.28–0.97	**0.027**
**CD206**						
Low	1			1		
High	0.54	0.35–0.83	**0.005**	0.51	0.31–0.82	**0.005**

**Table 4 cancers-14-04829-t004:** Multivariate analysis. The bold: the *p*-value was significant.

Variables	OS			RFS		
	HR	95% CI	*p*-Value	HR	95% CI	*p*-Value
**Tumor size**			**0.004**			
T1	1					
T2/T3/T4	2.03	1.23–3.34				
**Nodal status**			**<0.001**			**<0.001**
N−	1			1		
N+	2.57	1.65–4.00		4.87	2.91–8.12	
**Adjuvant chemotherapy**			**<0.001**			**0.004**
No	1			1		
Yes	0.34	0.22–0.53		0.48	0.29–0.80	
**Histology**			**0.002**			
Ductal	1					
Other	0.37	0.18–0.76				
**TILs**						**0.030**
≤5%	1		**0.028**	1		
>5%	0.59	0.36–0.96		0.45	0.22–0.93	
**CD206**						0.073
Low				1		
High				0.63	0.33–1.04	
**CD163**						
Low				1		**0.872**
High				1.07	0.49–2.34	

Variables with a *p*-value < 0.15 in univariate analysis were included in the multivariate model. Only significant variables are presented in the table, with the exception of CD206 and CD163 that were added for information despite the lack of statistically significant association.

## Data Availability

The datasets used and/or analyzed during the current study are available from the corresponding author on reasonable request.

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
