# Peer review of "Association of CD206 Protein Expression with Immune Infiltration and Prognosis in Patients with Triple-Negative Breast Cancer"

_cancers, 2022, doi:10.3390/cancers14194829_

Round 1

Reviewer 1 Report

Bobrie et al used immunohistochemistry to characterize the population of Tumor Associated Macrophages (TAMs) in a well characterized cohort of Triple Negative Breast Cancer (TNBC) patients. They applied four markers (including M1 and M2 specific markers) and investigated the association between their expression, clinicopathologic parametes and survival. This study highlights CD206 as a potential marker of interest. CD206, a marker of M2 macrophages, correlated with smaller size of tumors, number of Tumor Infiltrating Lymphocytes (TILs) and PD-L1 expression. Moreover, CD206 expression was shown to correlate with overall (OS) and relapse-free survival (RFS) in univariate analysis. On the other hand, multivariate analysis revealed no correlation between CD206 expression and OS and a trend for association of high CD206 expression with RFS, although not significant.

The study is very interesting and highlights a new subtype of TAMs (CD206+ TAMs) in TNBC that may have a favorable action and need to be characterized extensively in the future.

The paper is very well written, with attention to detail and a thorough description of the sound methodology. The results are clearly presented and the discussion is quite comprehensive.

Comments:

1. Given the data generated from the multivariate analysis, it is better to town down the conclusion of the paper, avoid the general term “ correlates with good prognosis” and stick more to the actual findings that convincingly demonstrate the potential favorable clinical outcome of the patients with TNBC enriched in CD206+ TAMs.

2. Have you attempted to stratify patients in M1 low/ M2 high and M1 high/ M2 low group based on the expression of IRF8 and CD206? I wonder if this would result in two groups with significantly different OS and RFS…

3. Although I appreciate the effort of the authors to explain thoroughly their methodology for defining the cutoffs/ high and low expression groups for statistical analysis, I think that it would be better to transfer bits of the discussion (for example lines 396-409) to the methods section.

4. Please clarify the meaning of lines 476-479

5. Jamiyan et al (2020) set the cutoff for low/high TILs at 30%. You used 5%. How did you decide this?

6. Line 109 Please explain the CIBERSORT acronym

7.Line 169 Please mention briefly the platform, antibody and method for immunohistochemical evaluation of PD-L1

8.Line 215 and line 346 Is the cutoff for multivariate analysis 0.15 or 0.05? If it is 0.15, then PD-L1 data should be included in Table 4

9.Line 245 Reference for Salgado guidelines is missing

10.Supplem figs 1D-2D Please correct typo IRF88

Reviewer 2 Report

Thank you for your work, however the paper has some flaws to be addressed:

- the manuscript should be revised by a native English speaker

- All the abbreviations used in the main text and in the figures\tables should be extensively clarified in a legend

- Figure 2 is blurred

Author Response

please see it in the attachment

Reviewer 3 Report

Dear Author/s,

Submitted manuscript Cancers-1843554 has done the good work on TNBC patient sample using simple immunohistochemistry method.

I would suggest title of the paper should be High infiltration of CD206+ M2-like macrophages in the tumor microenvironment indicates poor prognosis and survival in TNBC patients. But for that you have do present/modify your results.

Important finding/backbone of the presented work includes CD206 expression associated with immune infiltration and prognosis in TNBC patients, which should be shown in the figure/image form. Most of the data of the manuscript presented in a table format which is seems like disinteresting to the readers.

I would recommend authors can analyzed the multiple stained samples using digital imaging platform, so that cell features overlap down to a single-pixel level, using a CellProfiler v.2.1.1 pipeline, Alignment_Batch.cppipe (available under General Public License version 2 [GPLv2] at https://github.com/multiplexIHC/cppipe).

http://dx.doi.org/10.1016/j.celrep.2017.03.037

As author has mentioned that they are using 12-15 years old samples for experiments, I would suggest use the advance imaging analysis workflow and present the data in a summarized with an improved approach.

Is this statement being correct in the light of reference no. 25? You stated in Line no. 432-433 “CD206 is expressed in TAMs in several cancer types, but to our knowledge, this is the first report that analyzed the clinical relevance of CD206 TAMs in TNBC”

There is certain relevant literature has been not included in the discussion. Cells. 2021 Dec 2;10(12):3393. doi: 10.3390/cells10123393.

Overall, I would recommend improve the result section and present the data in advanced figure format.

All the best.

Author Response

please see it in the attachment

Round 2

Reviewer 3 Report

Authors has done some correction in the discussion and conclusion section but in the submitted revised version, I did not find any immunohistochemistry images in the main text file for TAM. 

Author Response

Dear reviewer,

As you advised, we included a figure presenting immunohistochemestry image of TAMs in the main text. 

We hope you will find this new figure informative.

Best regards,

Angélique Bobrie
